# Photoredox-catalyzed C–C bond cleavage of cyclopropanes for the formation of C($sp^3$)–heteroatom bonds

Liang Ge[1,2], Chi Zhang[1,2], Chengkai Pan[1], Ding-Xing Wang[1], Dong-Ying Liu[1], Zhi-Qiang Li[1], Pingkang Shen[1], Lifang Tian[1] & Chao Feng ®[1] ✉

Sterically congested C–O and C–N bonds are ubiquitous in natural products, pharmaceuticals, and bioactive compounds. However, the development of a general method for the efficient construction of those sterically demanding covalent bonds still remains a formidable challenge. Herein, a photoredox-driven ring-opening C($sp^3$)–heteroatom bond formation of arylcyclopropanes is presented, which enables the construction of structurally diversified while sterically congested dialkyl ether, alkyl ester, alcohol, amine, chloride/fluoride, azide and also thiocyanate derivatives. The selective single electron oxidation of aryl motif associated with the thermodynamic driving force from ring strain-release is the key for this transformation. By this synergistic activation mode, C–C bond cleavage of otherwise inert cyclopropane framework is successfully unlocked. Further mechanistic and computational studies disclose a complete stereoinversion upon nucleophilic attack, thus proving a concerted $S_N2$-type ring-opening functionalization manifold, while the regioselectivity is subjected to an orbital control scenario.

C($sp^3$)–heteroatom bond construction represents one of the fundamental operations which finds a wide application in pharmaceutical relevant research, material science and agrochemical development, among others[1]. In particular, the assembly of sterically congested ethers/amines are of intense interests. However, conventional methods to access these targets always faces daunting challenges[2]. While the venerable $S_N2$ reactions as exemplified by Williamson ether synthesis[3,4] and Mitsunobu reaction[5,6] are virtually straightforward, they are, however, heavily restricted to primary and secondary alkyl electrophiles. The synthetic elaboration of tertiary congener still remains problematic in modern synthetic organic chemistry because of the intrinsic reaction nature (Fig. 1a). On the other hand, the $S_N1$ reactions which traverse through carbocation species often lead to stereoablative transformation, although they are relatively insensitive to steric encumbrance because of the flattened reaction intermediates involved (Fig. 1a)[7–10]. More recently, by taking advantage of a

controlled generation of carbocation intermediate from electro- or photo-catalyzed decarboxylation, Baran[11] and Nagao/Ohmiya's[12,13] groups achieved efficient synthesis of sterically-hindered dialkyl ethers. Despite these advancements, the development of new strategies that encompass the merits of both $S_N2$ (for stereochemistry transfer) and $S_N1$ reactions (for steric insusceptibility) would be of undeniable significance.

Specifically, nucleophilic displacement for forging tert-C($sp^3$)–heteroatom linkage is made possible by either leveraging electronically activated substrates such as tertiary electrophiles bearing electron-withdrawing groups (the reaction mode varies depending on respective catalytic regimes, through either quasi-$S_N2$[14–18], or metal-promoted stereoconvergent coupling manifold[19–25]) or utilizing bespoke precursors for entropically more favored intramolecular annulations. For example, an elegant work from Cook and co-workers has revealed that intramolecular substitution of tertiary alcohol for

[1]Technical Institute of Fluorochemistry (TIF), Institute of Advanced Synthesis (IAS), School of Chemistry and Molecular Engineering, State Key Laboratory of Material-Oriented Chemical Engineering, Nanjing Tech University, 30 South Puzhu Road, Nanjing 211816, China. [2]These authors contributed equally: Liang Ge, Chi Zhang. ✉e-mail: iamcfeng@njtech.edu.cn

**a** Reaction profiles of $S_N2$ and $S_N1$ displacements

Restricted to 1° and 2° alkyl electrophiles Stereoablative transformation

**b** Stereoinvertive $S_N1$ displacement via tight-ion pairs

$FeCl_3$, $AgSbF_6$, $CH_2Cl_2$

shielded face

**c** Strain-release-driven stereoinvertive ring-opening substitution

Lewis or Brønsted acids

**d** SET-oxidation/strain-release promoted $S_N2$-type ring-opening functionalization

PC, $(PhS)_2$

DCE, 2,6-ditBuPy
Blue LED, $N_2$, rt

D-D Cyclopropane

SET

√ redox-neutral √ high regio/stereoselectivity
√ broad scope √ mild reaction conditions

HAT

remote activation

D-A Cyclopropane Benzylic radical

**Fig. 1 | Nucleophilic substitution at sterically hindered carbon centers. a** Elementary reaction profiles of $S_N2$ and $S_N1$ reaction. **b** Iron-catalyzed stereoinvertive intramolecular $S_N1$ displacement. **c** Lewis or Brønsted acid catalyzed stereoinvertive ring-opening substitution promoted by strain release. **d** This work.

cyclic sulfonamide synthesis could be readily accomplished by employing iron salt as Lewis acidic catalyst[26,27]. The chirality transfer of enantioenriched alcohols was rationalized by a stereoinvertive displacement involving tight-ion pairs (Fig. 1b). As an alternative, by harnessing ring-strain release[28–31], small heterocycles such as epoxide and aziridine could also readily participate in nucleophilic displacement of those strained C–N/O bond[32,33]. However, the ring-opening cleavage of C–C bond embedded in small carbocycles such as cyclopropane is relatively underexplored except for those electronically biased donor-acceptor ones[34–37]. Recently, a strain-release-driven stereoinvertive ring-opening substitution of cyclopropyl carbinol derivatives was accomplished by Marek and coworkers (Fig. 1c)[38]. Also of note, a pioneering work of ring-opening etherification of arylcyclopropane enabled by ultraviolet light sensitization was disclosed by Rao and Hixson[39]. Further mechanistic study by Dinnocenzo et al. confirmed the involvement of a cyclopropane radical cation intermediate and a three-electron $S_N2$-like pathway[40,41]. However, the harsh conditions and use of specific apparatus undoubtedly limit its broader synthetic application. In 2019, our group reported a visible light-promoted oxo-amination of aryl cyclopropanes using nitrogen-containing heterocycles as the nucleophiles under aerobic conditions[42]. By capitalizing on a similar ring-opening strategy, elegant works from Studer and co-workers further enabled 1,3-

difunctionalization of arylcyclopropanes under photoredox or cooperative NHC/photoredox catalysis[43,44]. In addition, by resorting to electrocatalysis, Werz et al. have realized a facile ring-opening functionalization of donor–acceptor cyclopropane/cyclobutane[45,46].

While notable progress has been attained during the past several years, the underlying problems such as harsh reaction conditions, limited substrate generality, employment of hazardous reagents are still of big concern. Therefore, the development of novel protocols that encompass wide substrate scope and mild reaction conditions without using strong base/acid, extra oxidant/reductant is still highly desirable. With our continuing interest in developing strategically novel synthetic transformation with photoredox catalysis[42,47–53], here we report a protocol for the construction of sterically congested $C(sp^3)$–heteroatom bonds via ring opening functionalization of electronically unbiased aryl cyclopropanes (Fig. 1d). Notable features of the present strategy include: i) selective SET-oxidation enables the smooth generation of aryl cation radial intermediate[54–63], which engenders a remote activation of cyclopropane skeleton through σ to SOMO orbital interaction[49,50,64]; ii) the synergistic substrate activation (via intentional generation of donor-acceptor cyclopropane cation radical intermediate[34–37]) and strain-release enables a stereoselective $S_N2$-like interaction with relatively weak nucleophile under essentially neutral reaction conditions; iii) the integration of SET process and

**Table 1 | Reaction conditions optimization**

| Entry | Catalyst | Yield (%) |
|---|---|---|
| 1 | PC-I | 38 |
| 2 | PC-II | 25 |
| 3 | PC-III | 47 |
| 4 | PC-IV | 40 |
| 5 | PC-V | 73 |
| 6 | PC-VI | 68 |
| 7 | PC-VII | 63 |
| 8 | PC-VIII | 86 (isolated yield) |
| 9 | PC-IX | NR |
| 10 | PC-VIII (5 mol%) | 91 |
| 11 | PC-VIII (without light or under air) | NR |
| 12 | - | NR |
| 13 | PC-VIII (with 10 eq. EtOH) | 92 (isolated yield) |

Experiments were performed with **1** (0.1 mmol), **H-Nu** (0.15 mmol), photocatalyst (2 mol%), 2,6-di-$^t$Bu-py (25 mol%), (PhS)$_2$ (20 mol%) in DCE (0.5 mL), irradiating with 15 W blue LED under N$_2$ atmosphere at room temperature for 48 h. Yield and conversion were determined by $^1$H NMR using 1,1,2,2-tetrachloroethane as internal standard. NR, no reaction. 2,6-di$^t$Bu-py, 2,6-di-tert-butylpyridine. DCE, 1,2-dichloroethane.

hydrogen atom transfer (HAT) event enables a redox-neutral transformation; iv) a LUMO orbital coefficient distribution controlled regioselectivity was proposed, which is supported by DFT calculation.

# Results

## Reaction optimization

Based on the cyclic voltammetry measurement of aryl cyclopropane substrate **1** ($E_{1/2}^{ox}$ = +1.30 V vs. saturated calomel electrode in MeCN)[42], the model reaction between **1** and trifluoromethanesulfonamide (TfNH$_2$) was attempted with selected photocatalysts (PC-I to PC-IX) under N$_2$ atmosphere. After extensive investigation of reaction parameters (selected conditions were listed in Table 1, for detailed condition optimization, see Supplementary Methods), we were pleased to find that under the irradiation of blue LEDs while using acridinium

series as photocatalyst and diphenyl disulfide as the hydrogen atom transfer (HAT) reagent, the model reaction conducted in DCE provided the desired product **2** in moderate to good yields (Table 1, entries 1–9)[65,66]. Among these, the PC-VIII ($E_{1/2}$(Acr*/Acr·) = +2.08 V vs. SCE)[67] proved to be the best, which afforded product **2** in 86% yield (entries 8). Further fine tuning of the reaction parameters eventually led to the formation of product **2** in 91% yield (entries 10). Negative control experiments clearly demonstrated the critical role of photocatalyst, light irradiation and nitrogen atmosphere in the present transformation (entries 11–12). Pleasingly, the optimal reaction condition for amination was also well suited for sterically hindered ether construction simply using alcohol as the nucleophile, and in the case of ethyloxylation reaction the desired product **3** could be readily obtained in 92% yield (entries 13). It is worth pointing out that all these nucleophilic

ring-opening reactions occurred in a highly regioselective manner without any regioisomers derived from nucleophilic attack at methylene or methine carbon atoms being observed.

## Examination of substrate scope

With the optimized reaction conditions in hand, the substrate scope with respect to arylcyclopropane was investigated using ethanol, TfNH₂ and pyrazole as nucleophiles (Fig. 2). It was found that cyclopropanes with electron-donating groups on the phenyl ring, such as methoxyl, alkyl, and phenyl reacted smoothly and provided the ethyl ethers (**3–5**, **7**), amides (**2, 28, 30, 32**) and α-quaternary substituted heterocycles (**27, 29**) in good yields, respectively. Moreover, polycyclic aromatic cyclopropane was also viable substrate as showcased by examples of **8** and **33**, though somewhat diminished reaction efficiency was observed in the latter case. When using unsubstituted phenyl cyclopropane with higher oxidation potential, the desired products could still be obtained in acceptable yields (**6, 31**), albeit requiring prolonged reaction time. Then the influence of the alkyl substituents on cyclopropyl ring system was examined. Changing the gem-dimethyl into bulkier gem-diethyl group had no obvious impediment on this reaction, and the etherification product **9** could be obtained in 95% yield, whereas 4-methoxyphenyl cyclopropane without alkyl substituent reacted sluggishly, leading to 26% yield of **37** in amination reaction. These contrasting experiments indicated that the steric demanding substituents on cyclopropane are not detrimental but conducive to the ring-opening cleavage, which could be rationalized by more effective positive charge delocalization as well as thermodynamically more favored ring-strain release. Spirocyclic substrates with differing ring size were proved amenable to this reaction, providing the desired cycloalkyl ether (**10–12**) and amine (**34–36**) derivatives in good yields. Furthermore, 1,2-arylalkyl-substituted cyclopropanes were also viable substrates irrespective of the stereochemistry, and remote functional groups, such as ester, aryl were well tolerated (**38, 13–14**). In these cases, excellent regioselectivities were still observed with nucleophiles attacking the alkyl-substituent carbon atom. It is also of note, cyclopropylamine derivatives were enabling substrates, which participated in ring-opening substitution reactions uneventfully to afford respective aminal products (**15–17, 39–41**). Additionally, 1,2-diaryl cyclopropanes were also revealed to be applicable (**18–20, 42–45**). When unsymmetric substrates were employed, the nucleophilic attack happened on the carbon atom containing more electron-rich aryl substituent with high to excellent regioselectivity (**19, 20, 43–45**). This unusual selectivity was further interpreted by DFT calculation (vide infra). The same principle was also applicable to the 1,2,3-triaryl substituted substrate, which secured a highly selective transformation (**25**). In the case of benzo-fused tricyclic substrates, the reaction occurred smoothly to generate ring-expansion products (**21, 22, 46, 47**), whereas for aryl-substituted bicyclic substrates the reaction happened selectively to afford trans-products **23, 48**, thus indicating a S_N2-like concerted nucleophilic attack/ring-opening mechanism (vide infra). Interestingly, 1,1-diaryl substituted cyclopropane was also competent substrate, which delivered the desired product **24** in moderate yield. Finally, an intramolecular ring-opening cyclization took place readily under the standard conditions to afford tetrahydrofuran derivative **26** in 78% yield.

Subsequently, the scope of nucleophile was explored. As for the oxygen-based nucleophile, a panel of structurally diversified alcohols containing different functional groups, differing in steric environment was tested first (Fig. 3). To our delight, primary alcohols such as methanol (**49**), (hetero)benzylic alcohol (**54–56, 62–64**), secondary alcohol such as isopropanol (**50**), 2-adamantanol (**72**) was applicable, delivering the desired products in moderate to good yields. Unfortunately, tertiary alcohol is not amenable under the current conditions, perhaps due to its low nucleophilicity and steric hindrance. Of note, water was proved to be the competent nucleophile which directly

offered tertiary alcohol product **57–59** in 61–77% yield. Furthermore, alcohols with various functional groups such as silyl ether (**51, 52**), Boc-protected amino group (**53**), terminal (**61**) or internal alkene (**60**) also readily engaged in this reaction, furnishing the desired products in moderate to good yields. To further showcase the synthetic applicability of this protocol, late-stage elaboration of a repertoire of natural product or pharmaceutical relevant alcohols were interrogated, which delivered respective alkylation products in synthetically useful yields (**65–67, 69–71, 73**). Finally, when 1,5-pentanediol was employed, a twofold functionalization was observed, giving rise to the product **68** in 61% yield.

Aside from alcohol, carboxylic acid also turned out to be competent O-type nucleophile, which provides an effective route for the ready access of sterically congested alkyl ester (Fig. 4). It is worthy of note that Baran's technic for decarboxylative ether synthesis could not be readily extrapolated for alkyl ester construction by leveraging cross-coupling of two different carboxylic acid molecules through selective mono-decarboxylation. During the course of substrate exploration, we found a wide spectrum of functionalities were well tolerated and carboxylic acids with diverse structure motifs engaged in this reaction. Aliphatic (**74–76**) and aromatic carboxylic acids (**77**) reacted with arylcyclopropanes with ease to produce the desired products in good yields. Despite the radical nature of reaction, the employment of unsaturated carboxylic acids (**78–80**) were also rewarding, without any interference on the reaction progress. Intriguingly, when using allylsulfone-derived carboxylic acid, the reaction proceeded smoothly to afford the desired product **80** in 74% yield and no radical-type desulfonylative annulation product was detected probably because of the kinetically unfavored 7-endo-trig radical cyclization as compared with HAT quenching process in this specific example. Furthermore, pharmaceutical (**81–84, 86**) and natural product (**85**) relevant carboxylic acids were also well accommodated in this reaction, showcasing the potential applicability of the present method. In accordance with intramolecular etherification, a carboxylic acid tethered cyclopropane delivered lactone **87** in excellent yield via an intramolecular ring-opening substitution process.

The generality of nitrogen-based heterocycle and other nucleophile was also briefly examined (Fig. 5). Pyrazole derivatives with 4-substituents such as Cl, Br, Me and 3-TBS derivative, as well as 1H-indazole, were all found to take part in this reaction smoothly and afforded the desired products in moderate to good yields (**88–92**). In addition, triazole derivatives were also applicable, though they are inclined to trigger the formation of regioisomeric products (**93, 94** and **95, 96**). Moreover, trimethylsilyl azide and trimethylsilyl isothiocyanate as proton-free nucleophilic reagents also turned to be competent under current conditions, affording respective products **97** and **98** in high yields with stoichiometric amount of tertiary alcohol as the hydrogen source (see Supplementary Methods for more details). The synthetically useful azide and thiocyano group provide versatile handles for divergent follow-up transformations.

Upon further exploration, we found that the applicable nucleophile was not restricted to oxygen or nitrogen-based entities, chlorination and fluorination could occur readily by using 2,6-lutidine hydrochloride and triethylamine trihydrofluoride as nucleophiles, respectively. For arylcyclopropane without alkyl substituent and symmetric 1,2-aryl cyclopropane, the 1,3-hydrochlorination reaction delivered the desired products in excellent yields (**100, 101**). Interestingly, when 1,2-arylalkyl cyclopropane was employed, two regioisomeric chlorination products were observed in excellent yields but low regioselectivity (**104:105** = 3.3:1, 96% and **106:107** = 2.3:1, 95%), which is in stark contrast to the aforementioned C–O and C–N bond formation reactions. In comparison, when using gem-dimethyl substituted aryl cyclopropane the reaction took place in high efficiency and regioselectivity (**108, 99**), whereas in the former case the obtained

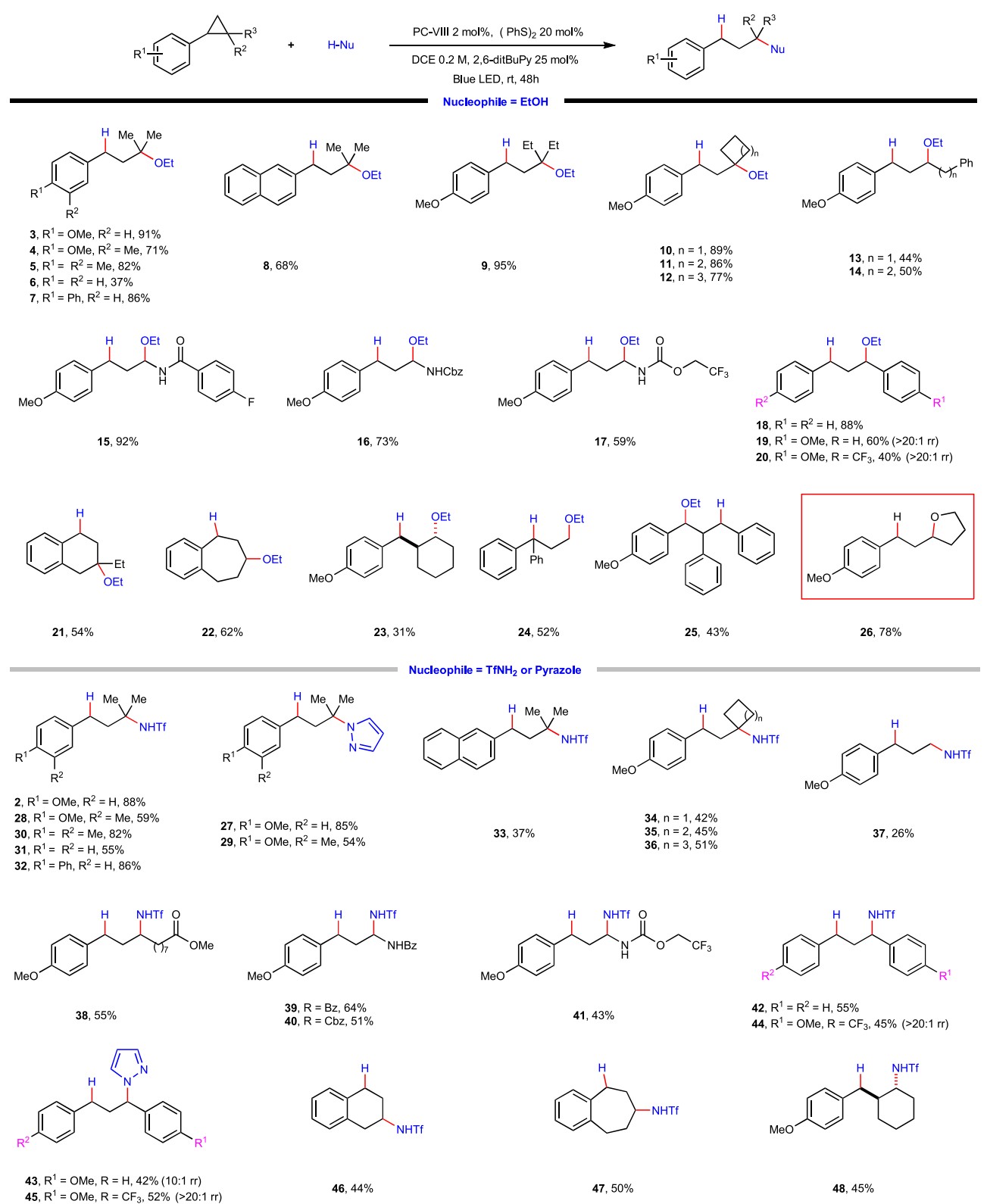

**Fig. 2 | Cyclopropane substrate scope of photoredox-coupled C(sp³)–heteroatom bond formation.** Reaction conditions: arylcyclopropane (0.2 mmol, 1 equiv.), EtOH (2 mmol, 10 equiv.) or TfNH₂ (0.3 mmol, 1.5 equiv.), photocatalyst (2 mol%), (PhS)₂ (0.04 mmol, 0.2 equiv.), 2,6-ditBu-Py (0.05 mmol, 0.25 equiv.) and DCE (1.0 mL, 0.2 M) at 25 °C for 48 h under irradiation of a 15 W blue LED lamp.

product was not stable, which underwent a spontaneous elimination reaction to afford isomeric mixture of alkene products (**109**:**110** = 2:1). In contrast, the corresponding fluorinated products were quite stable and **102**, **103** were obtained in good yields.

## Mechanistic investigations

In order to gain a deep insight into the mechanism, a set of control experiments were then executed. Firstly, a light "on–off" experiment demonstrated the necessity of continuous irradiation for the progress

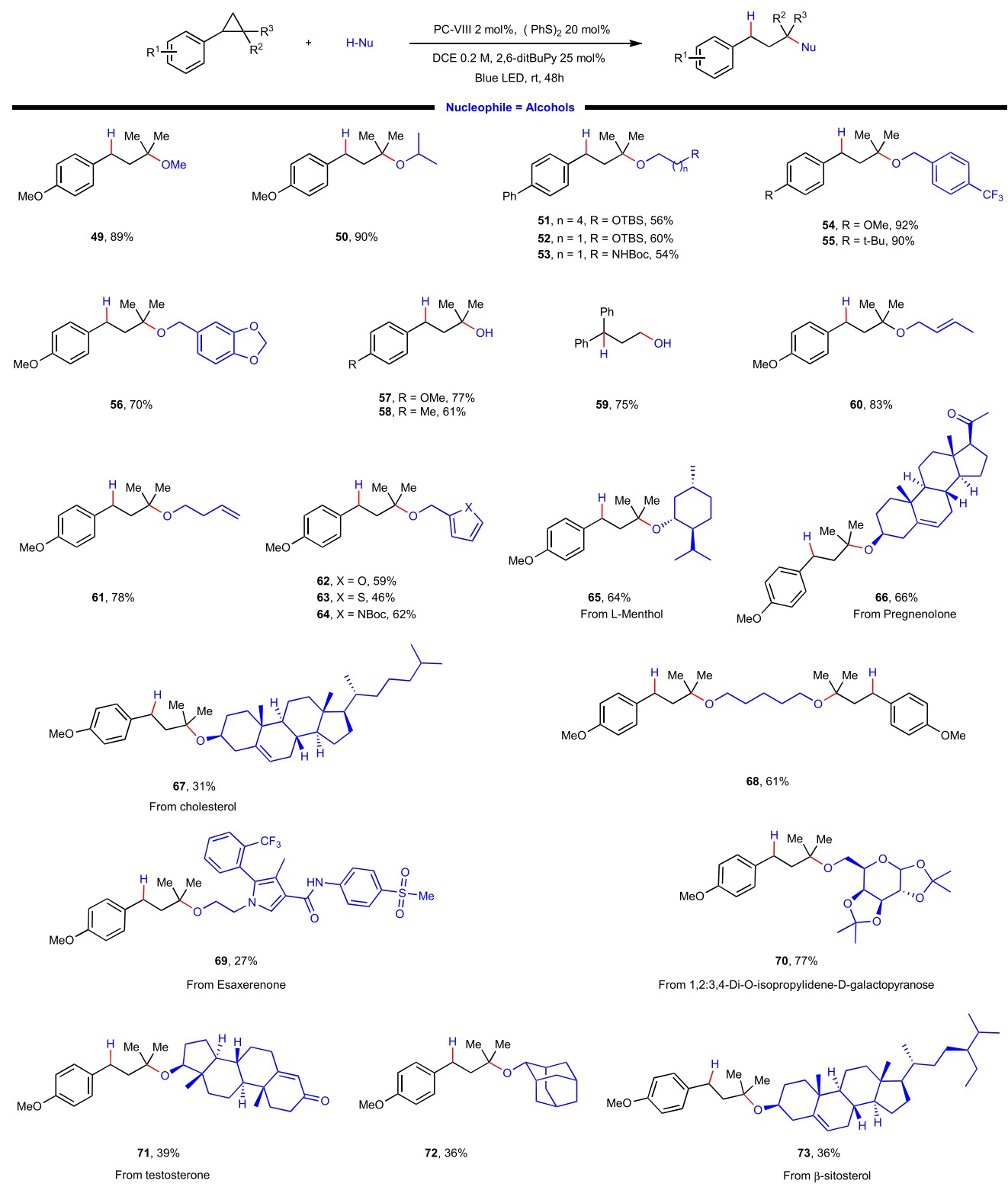

**Fig. 3 | Substrate scope of photoredox-coupled C($sp^3$)−heteroatom bond formation with alcohol as nucleophile.** Reaction conditions: arylcyclopropane (0.2 mmol, 1 equiv.), alcohol (0.4 mmol, 2 equiv.), photocatalyst (2 mol%), (PhS)$_2$ (0.04 mmol, 0.2 equiv.), 2,6-ditBu-Py (0.05 mmol, 0.25 equiv.) and DCE (1.0 mL, 0.2 M) at 25 °C for 48 h under irradiation of a 15 W blue LED lamp. **57−59**, H$_2$O (2 mmol, 10 equiv.) and dioxane (1.0 mL, 0.2 M) was used as nucleophile and solvent, respectively.

of this transformation. Secondly, addition of stoichiometric amount of TEMPO as radical inhibitor into the reaction of **1** and ethanol led to complete inhibition of product formation (Fig. 6a). Moreover, a radical clock experiment with **111** was conducted, which uneventfully delivered a cascade ring-opening product **112** in 88% yield, thus substantiating the involvement of benzylic radical intermediate in this reaction (Fig. 6b). Furthermore, the quantum yield of reaction between **1** and ethanol was determined to be 0.182, indicating that radical chain

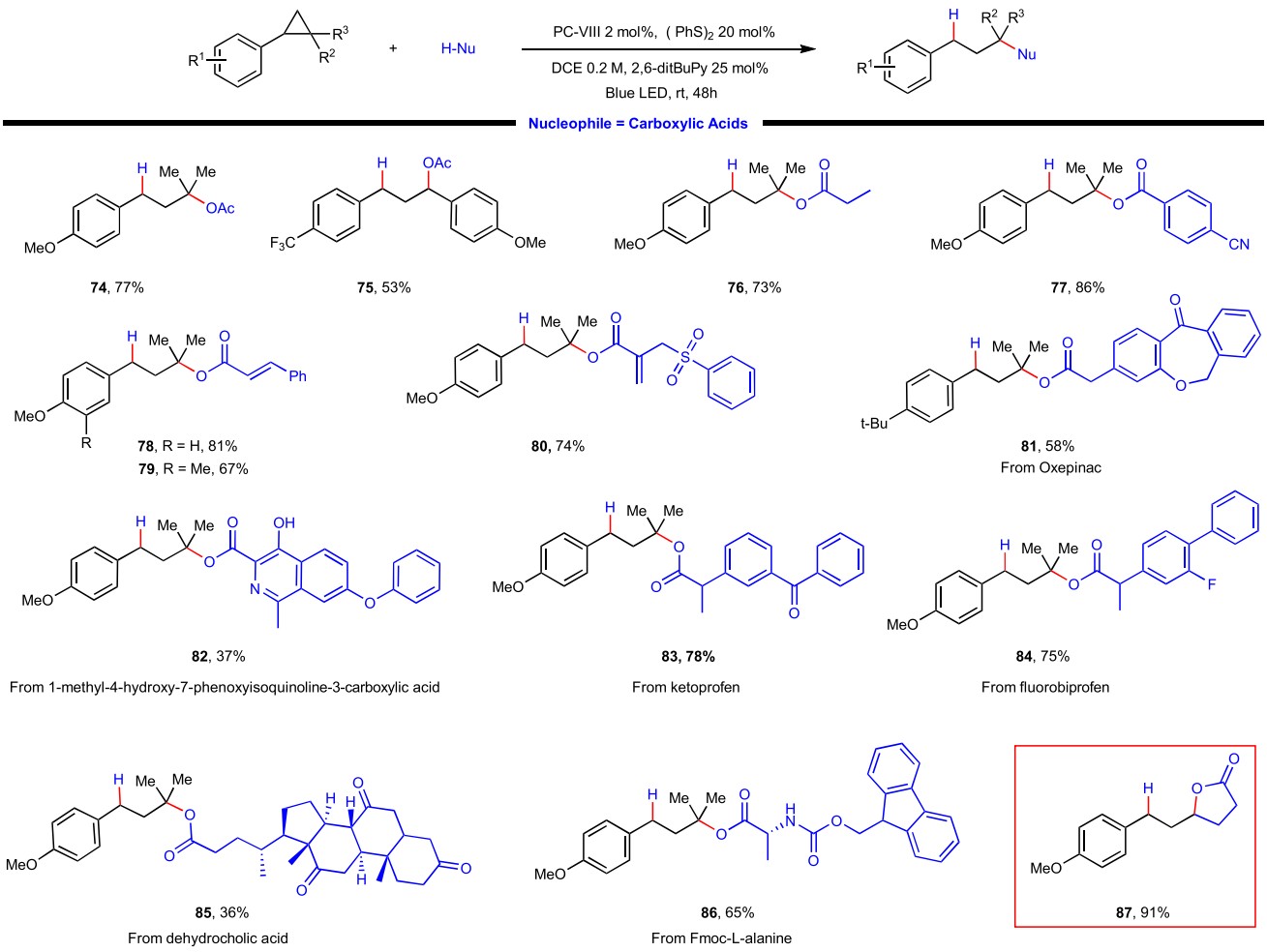

**Fig. 4 | Substrate scope of photoredox-coupled C($sp^3$)–heteroatom bond formation with carboxylic acid as nucleophile.** Reaction conditions: arylcyclopropane (0.2 mmol, 1 equiv.), carboxylic acid (2 equiv.), photocatalyst (2 mol%), (PhS)$_2$ (0.04 mmol, 0.2 equiv.), 2,6-ditBu-Py (0.05 mmol, 0.25 equiv.) and DCE (1.0 mL, 0.2 M) at 25 °C for 48 h under irradiation of a 15 W blue LED lamp.

process may not be the major pathway (see Supplementary Discussion for more details).

To further clarify the source of hydrogen at benzylic site of the product, control experiments were performed under standard conditions while using CDCl$_3$ as the reaction solvent, D4-methanol and methanol as respective nucleophiles (Fig. 6c). Though both reactions gave corresponding products in high yields (**113** and **114**), only reaction with D4-methanol led to the incorporation of considerable amount of deuterium at benzylic position (**113**). This result clearly demonstrated that the proton of the nucleophile, other than the solvent, was transferred to the benzylic site of the product.

In our previous work[42], the cyclopropane ring-opening step has been proved to take place through a concerted nucleophilic attack/ring-opening S$_N$2-like manner. To determine whether concerted nucleophilic S$_N$2 ring-opening functionalization is also involved in the present reaction, control experiment with enantiomerically enriched trans-1,2-diphenyl cyclopropane[42] **114** (90% ee) was examined with ethanol under the standard reaction conditions, which resulted in the generation of product **18** with 85% ee, thus unequivocally substantiate a concerted S$_N$2-like process with complete stereoinversion (Fig. 7, eq. a). The somewhat erosion of stereochemistry was rationalized as resulting from minor contribution of substrate racemization. Such stereochemistry scrambling could occur through either triplet energy transfer or SET-induced reversible ring-cleavage of cyclopropane substrate (see Supplementary

Discussion for more details). To further probe the generality of the stereoinvertive ring-opening substitution, more enantioenriched structurally differing substrates were tested. A 1,2-diaryl cyclopropane **115** with an electron-deficient benzothiazole also underwent the reaction smoothly to afford enantiopurity conserved product **116** in high yield and excellent regioselectivity (Fig. 7, eq. b). Furthermore, under standard conditions, with 2,6-lutidine hydrochloride as nucleophile, an enantioenriched 1,2-arylalkyl-substituted cyclopropane **117** with a naked hydroxyl group underwent chemoselective chloride attack, without intramolecular epoxidation or intermolecular ether formation being observed. However, as in the case of **104** and **106**, two regioisomeric chlorination products were observed, with enantiopurity fully conserved product **118** being the minor isomer (Fig. 7, eq. c). The unique reactivity of chloride was intriguing as previous mechanistic study and other cases in this work all showed a high regioselectivity for nucleophilic attack at the more substituted carbon atom[50]. We ascribed this unusual selectivity to the high nucleophilicity of naked chloride ion thus resulting in relatively small difference of energy barriers for site discrimination.

During the investigation, the apparently abnormal regioselectivities in some cases have attracted our attention (**19**, **20**, **43**, **44**, **45**, **116**). Aiming to provide an explanation for the regioselectivity of these 1,2-diaryl cyclopropanes, we have conducted DFT calculation to study the frontier molecular orbitals of cation radical of some representative

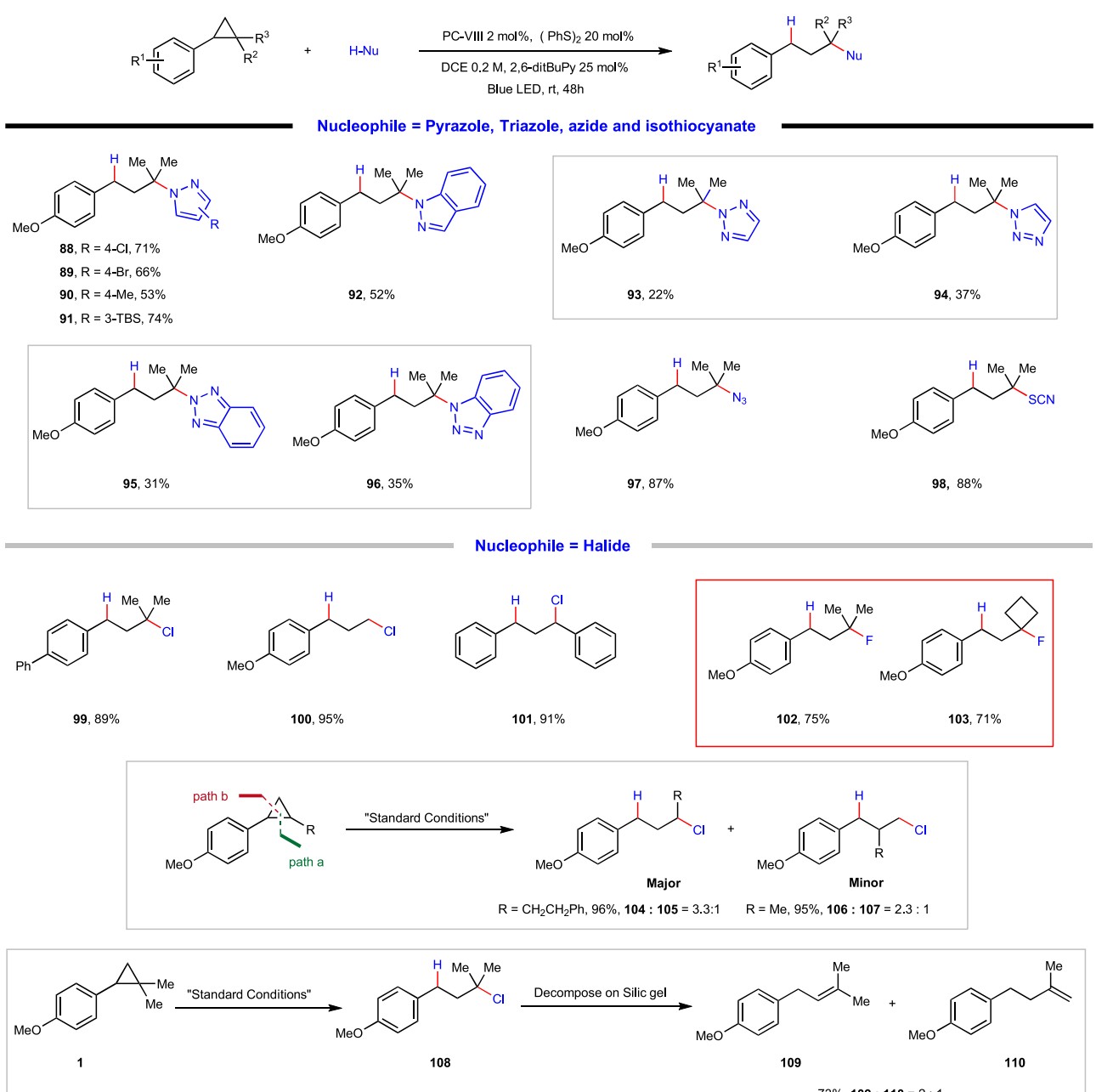

**Fig. 5 | Substrate scope of photoredox-coupled C($sp^3$)–heteroatom bond formation with other hetero-nucleophiles.** 88–96: arylcyclopropane (0.2 mmol, 1 equiv.), heterocycle (1.5 equiv.), photocatalyst (2 mol%), (PhS)₂ (0.04 mmol, 0.2 equiv.), 2,6-ditBu-Py (0.05 mmol, 0.25 equiv.) and DCE (1.0 mL, 0.2 M) at 25 °C for 48 h under irradiation of a 15 W blue LED lamp. **97**–**98**: arylcyclopropane (0.2 mmol, 1 equiv.), TMSN₃/TMSNCS (2.0 equiv.), 2-methyl-2-butanol (0.6 mmol, 3.0 equiv.), photocatalyst (2 mol%), (PhS)₂ (0.04 mmol, 0.2 equiv.), 2,6-ditBu-Py (0.05 mmol, 0.25 equiv.) and DCE (1.0 mL, 0.2 M) at 25 °C for 48 h under irradiation of a 15 W blue LED lamp. **99**– **110**: arylcyclopropane (0.2 mmol, 1 equiv.), 2,6-lutidine hydrochloride/triethylamine trihydrofluoride (2.0 equiv.), photocatalyst (2 mol%), (PhS)₂ (0.04 mmol, 0.2 equiv.), 2,6-ditBu-Py (0.05 mmol, 0.25 equiv.) and DCE (1.0 mL, 0.2 M) at 25 °C for 48 h under irradiation of a 15 W blue LED lamp. TMSN₃, azidotrimethylsilane. TMSNCS, trimethylsilylisothiocyanate.

substrates. The DFT calculations were performed with the Gaussian 09 program. Geometries of the minimum energy structures were optimized at the B3-LYP level of theory with the 6-31G(d, p) basis. As shown in Fig. 8, the highest contribution of LUMO orbital is located at the α position of the more electron-rich aryl ring (C3 site of **120-Int** and C2 site of **115-Int**), which is also the reaction site of nucleophilic substitution. Furthermore, the frontier molecular orbital of radical cation of model substrate **1-Int** was calculated and the result indicates the highest LUMO orbital coefficient at the dialkyl substituted carbon. Taken together, we believe that regioselectivity of this three-electron

$S_N2$ reaction was subjected to an orbital control scenario whereby the site with higher LUMO orbital coefficient is more prone to be attacked by external nucleophiles (see Supplementary Discussion for more details of the DFT calculation).

Based on the above experimental results, a plausible mechanism was proposed (Fig. 9). Initially, the excited-state of acridinium catalyst was reached by visible light excitation (450 nm). Then reductive quenching of the excited photocatalyst by aryl cyclopropane produced arylcyclopropyl cation radical[58,59] intermediate **I**, which characteristically resembles the donor-acceptor cyclopropane[37–42], and

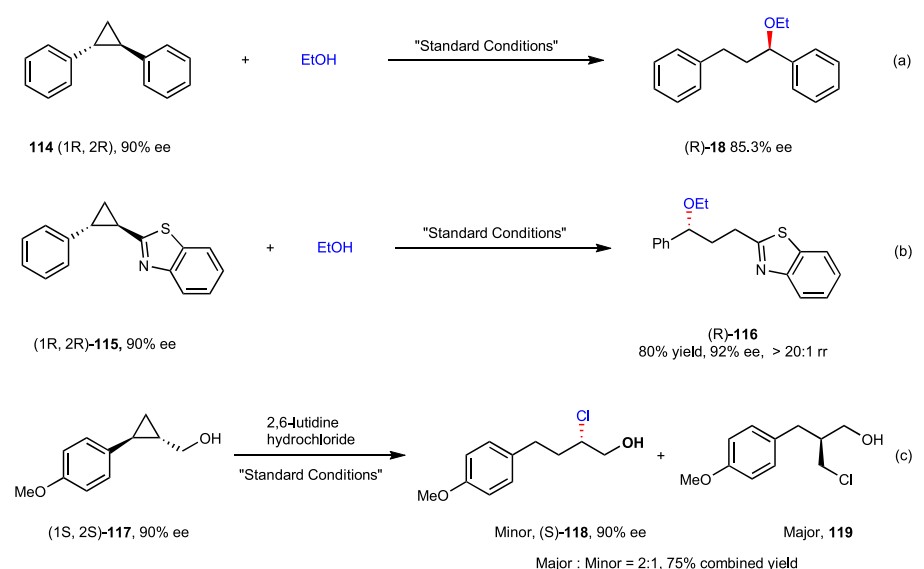

**Fig. 6 | Mechanistic Investigations. a** Experiment with radical inhibitor. **b** Radical clock experiment. **c** Experiments probing the hydrogen source at benzylic site of the product.

**Fig. 7 | Stereospecific reaction with enantioenriched substrates. a** Reaction of enantioenriched trans-1,2-diphenyl cyclopropane **114** with ethanol. **b** Reaction of enantioenriched 1,2-di(hetero)aryl cyclopropane **115** with ethanol. **c** Reaction of enantioenriched 1-aryl-cyclopropyl-carbinol 117 with 2,6-lutidine hydrochloride.

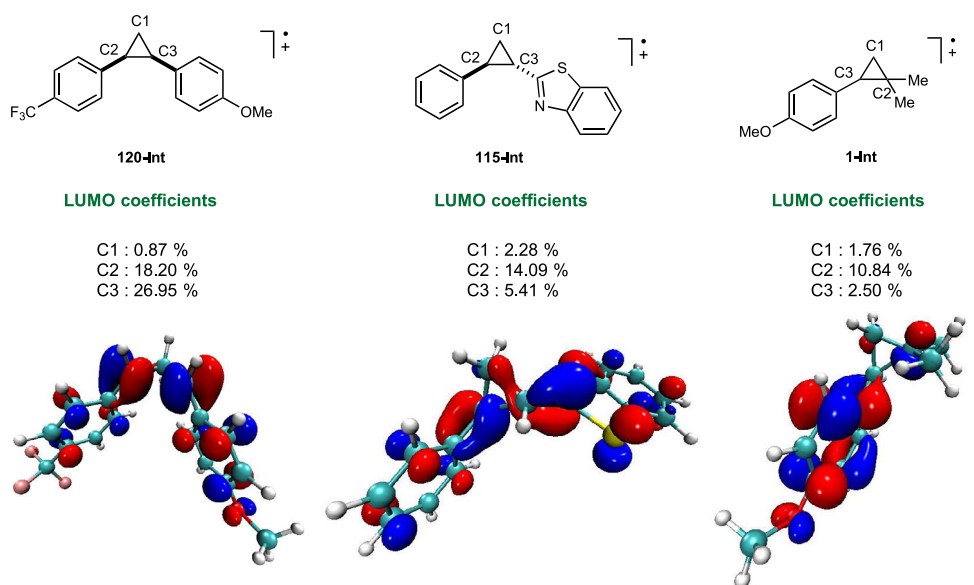

**Fig. 8 | Frontier molecular orbitals of representative substrates.** Schematic representation of **120-int**, **115-int**, and **1-int** (top) along with the representation of their frontier molecular orbitals (bottom). See Supplementary Information for more details of the DFT calculations.

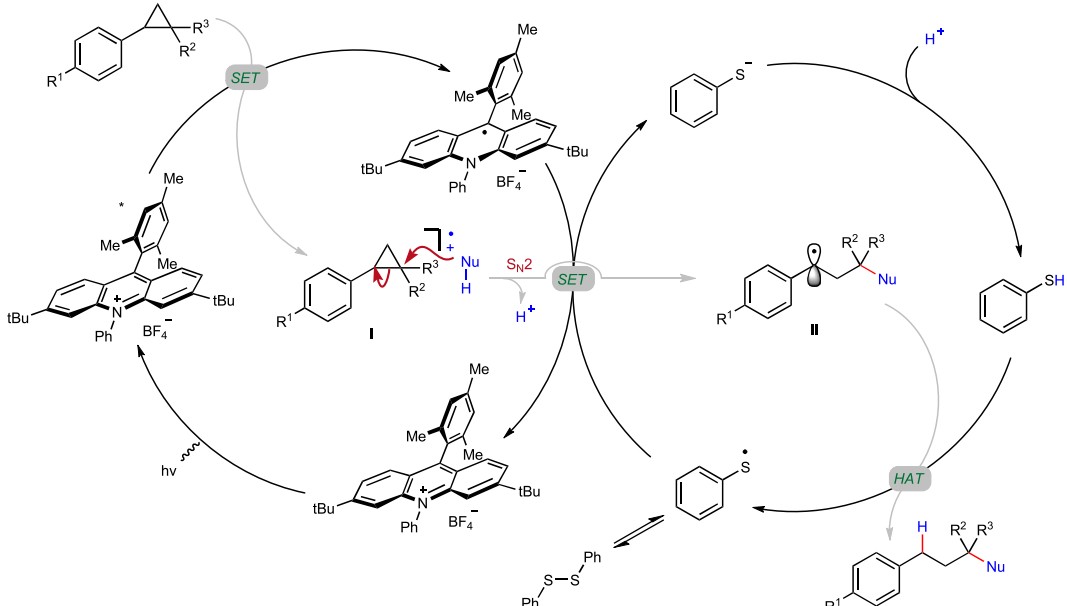

**Fig. 9 | Proposed reaction mechanism.** Key step: regio- and stereoselective nucleophilic ring-opening attack of arylcyclopropane cation radical intermediate by external hetero-nucleophile for sterically congested C($sp^3$)–heteroatom bond formation.

acridinium radical as one-electron-reduced catalyst. The subsequent nucleophilic attack of nucleophile selectively afforded ring-opened carbon centered radical intermediate **II**. The target product was finally produced via hydrogen atom transfer (HAT) from the thiol, accompanied by the generation of thiol radical. The active HAT agent was regenerated via reduction of the thiol radical by acridinium radical associated with subsequent proton transfer[68], meanwhile closing the photocatalysis cycle.

## Discussion

In summary, by taking the advantage of synergistic effect of photocatalyzed SET-oxidation and strain-release, we have developed a protocol for sterically congested C($sp^3$)–heteroatom bond construction via ring-opening cleavage of C–C bond of electronically unbiased arylcyclopropanes. The reaction features a wide applicability to different kinds of nucleophiles, including alcohol, carboxylic acid, sulfonamide, nitrogen-containing heteroaromatic, chloride and even water, excellent regioselectivity, mild reaction conditions, redox-neutral process. Wide substrate scopes with respect to both cyclopropane and nucleophile are delineated and late-stage functionalization of an array of natural product and pharmaceutical molecule derivates are also presented. Mechanistic studies prove a $S_N2$-like ring-opening mechanism, which provides opportunity for the construction of synthetically challenging chiral quaternary stereocenters with enantiometrically enriched substrates. More diverse patterns of ring-opening functionalization of arylcyclopropane are under way in our laboratory.

## Methods

### General procedure for the C−O bond formation with alcohol as nucleophile

In a typical experiment, to a mixture of (PhS)$_2$ (9.0 mg, 0.04 mmol), PC-VIII (2.3 mg, 0.004 mmol) and propan-2-ol (24 mg, 0.4 mmol) in 1,2-dichloroethane (1.0 mL) was added 2,6-di-$^t$Bu-Py (9.6 mg, 0.05 mmol) and arylcyclopropane **1** (0.2 mmol) under nitrogen atmosphere. After 48 h irradiation with a 15 W blue LED lamp (l = 459 nm) at room temperature, the mixture was evaporated to dryness under reduced pressure, and the resulting residue was purified by column chromatography on silica gel to afford the desired product **50** (42.5 mg, 90%). The reaction with other kind of nucleophile were carried out similarly and the procedures are presented in Supplementary Methods.

## Data availability

Detailed experimental procedures and characterization of new compounds are available in Supplementary Information. For condition optimization and experimental procedures, see Supplementary Methods. For mechanistic studies, spectroscopic data, HPLC and NMR spectra of compounds, see Supplementary Discussion. Further relevant data are available from the corresponding author upon request.

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

## Acknowledgements
We gratefully acknowledge the financial support of the National Natural Science Foundation of China (grant no. 21871138, 22271151), the Distinguished Youth Foundation of Jiangsu Province, and the Natural Science Foundation of Jiangsu Province (grant no. BK20220327).

## Author contributions
C.F. directed the investigations. C.F. and C.Z. prepared the manuscript. L.G., C.Z, C.P., D.W., D.L., Z.L., and P.S. performed the synthetic experiments and analyzed the experimental data. L.T. performed the DFT calculations.

## Competing interests
The authors declare no competing interests.
