## [Peer Review File · Nature Communications]

Photoredox-catalyzed C–C bond cleavage of cyclopropanes for the formation of C(sp³)–heteroatom bondsREVIEWER COMMENTS

Reviewer #1 (Remarks to the Author):

This paper by Chao Feng and co-workers submitted to Nature Communications reports on the ring-opening of aryl cyclopropanes using a photocatalytic system. Numerous nucleophiles including alcohols, nitrogen-based nucleophiles, and (pseudo) halides are added in a 1,3-fashion. The proton is located next to the aryl residue. Moderate to good yields have been obtained. In contrast to common donor-acceptor cyclopropane chemistry the nucleophiles does not attack at the carbon next to the aryl, but at the other. A plausible mechanism based on the most reasonable experiments is provided. In general, this is a very nice report that heavily expands the ring-opening chemistry of cyclopropanes. I strongly support its publication in Nature Communications after some minor revisions noted below:

- 1) The title is a bit misleading. I think the wording "sterically congested" is not necessary. However, I would definitely put "cyclopropane" in. What about "Photoredox-Catalyzed C-C Bond Cleavage of Cyclopropanes for the Formation of C(sp³)-Heteroatom Bonds"?
- 2) Although the introduction provides a broad overview of different methods, recent work on the electrochemically triggered ring-opening of cyclopropanes is missing. A sentence should be added, the respective references should be included: *Angew. Chem. Int. Ed.* 2021, 60, 15928; *Org. Lett.* 2021, 23, 5549.
- 3) From my point of view the numbering/nomenclature of the compounds is very confusing. Why not using simple numbers and a,b, c etc.
- 4) Why do the authors state that their reactions schemes are figures? I think Scheme would be more appropriate.
- 5) *endo* should be written in italics.
- 6) From my point of view it is not clear whether 3aS1 is a thiocyanate (RS-CN) or an isothiocyanate (R-N=C=S). Were the reported data compared with literature-known ones? It should be clearly elucidated which part of the amphiphilic nucleophile attacks.
- 7) Mechanism: Are there any ideas of which is the rate-limiting step? An in-depth comparison on the kinetics with different cyclopropanes (as it was once made for donor-acceptor cyclopropanes: *Angew. Chem. Int. Ed.* 2019, 58, 1955) was not made and would exceed the scope of this study. Nevertheless, is there any information whether the kinetics is more determined by the type of cyclopropane or the type of nucleophile?
- 8) Supporting Information: From my point of view, IR data are an important part of characterizing new compounds. These data should be added. I guess that also the question whether it is RSCN or RCNS might be easily solved with IR spectroscopy.

Reviewer #2 (Remarks to the Author):

The manuscript by Feng and co-workers describes a photoredox/HAT catalysed hydrofunctionalisation of aryl cyclopropanes with a diverse range of heteroatom nucleophiles. Key to the success of the reaction is the single-electron oxidation of the aryl cyclopropanes into radical cations, which converts the cyclopropane into an electrophilic species that more closely matches the reactivity of donor-acceptor cyclopropanes. The authors have previously reported related nucleophilic ring-opening reactions of aryl cyclopropane radical cations (see references 47 and 53), however, in the current manuscript they demonstrate that the benzylic radical intermediates can be intercepted by a hydrogen atom transfer (HAT) with thiophenol, instead of trapping with oxygen or allyl sulfones. An impressive substrate scope is demonstrated for both the aryl cyclopropanes and nucleophiles, which include alcohols, triflamide, N-heterocycles, chloride and fluoride. The yields are generally high and the regioselectivities are excellent in nearly all cases. The authors also provide DFT calculations to explain the high regioselectivity observed for unsymmetrical 1,2-diarylcyclopropanes.

Although this is a nice piece of work, I do not think it represents a significant advance to the field. The methodology is a logical extension to the author's (references 47 and 53), and Studer's (references

56-57), previous work where they have replaced various radical traps with diphenyl disulfide to enable HAT. It appears that the authors have simply taken Nicewicz's previously reported conditions and substrate scope for hydrofunctionalisations of aryl alkenes (reference 58; J. Am. Chem. Soc. 2012, 134, 18577; J. Am. Chem. Soc. 2013, 135, 9588; J. Am. Chem. Soc. 2013, 135, 10334; Angew. Chem. Int. Ed. 2014, 53, 6198; Nat. Chem. 2014, 6, 720) and applied them to aryl cyclopropanes. Indeed, the authors have already demonstrated that the transition from aryl alkenes (reference 49) to aryl cyclopropanes (reference 53) is readily achievable in fluoroallylation reactions. In addition, little further insight into the mechanism of the reaction has been provided (see reference 45-47 and 56-57).

Based on this, I do not support publication in Nature Communications. The authors should consider the following points before resubmission to another journal:

- The manuscript would benefit from careful proofreading, as there are numerous grammatical errors.
- Error with H/R3 groups in Figure 1a.
- In Figure 1d, the reaction arrow showing the "intramolecular electron shift" structure to the "benzylic radical" is misleading because these are not separate species, they are resonance structures.
- $E_{0/1/2}$ should be defined or rewritten as $E_{1/2}$ or E_{ox} to avoid confusion with standard reduction potential (E_{ox}).
- Line 109: "Diphenyl sulfide" should be "diphenyl disulfide"
- The hydroaminations reported by Nicewicz (J. Am. Chem. Soc. 2013, 135, 9588; Angew. Chem. Int. Ed. 2014, 53, 6198) need to be cited because it appears that these are where the authors took their initial reaction conditions from.
- Tertiary alcohols are a notable omission from the nucleophile scope. These should be investigated and the results included, even if the reactions fails.
- The failed reaction in the presence of TEMPO in Figure 2a is not necessarily indicative of a radical-engaged reaction. It likely that TEMPO simply reductively quenches the excited state photocatalyst, which prevents the formation of any alkyl radical intermediates, rather than functioning as a radical trap. Proof of a radical-engaged reaction would require the observation of a alkyl radical-TEMPO adduct.
- Line 264: "During the investigation, the unusual regioselectivities in some cases have attracted our attention (3tO1, 3uO1, 3tN2, 3uN1, 3uN2, 3adO1) – should it be 3ahO1?"
- Line 265: "Results of previous mechanistic studies that the nucleophilic substitution tends to produce the more stable radical intermediate, i.e. substitution occur at the β position of the electron-rich aryl ring, was in contradiction with the reaction outcome." – the reaction outcomes do not contradict this because benzylic radicals are always generated. Also, this statement is not relevant to the calculations in figure 4, which only include the LUMO coefficients of the radical cation, not the stability of the radical formed after nucleophilic attack.
- Line 282: "acridine photocatalyst" should be changes to "acridinium photocatalyst".
- The title of reference 15 is wrong.

Signed: A Noble

Reviewer #3 (Remarks to the Author):

The authors present a novel type of photoredox-coupled C(sp³)-heteroatom bond formation reaction, which involves a single electron transfer oxidation of the aryl group and C-C bond cleavage. Although the involvement of iridium may not be a desirable feature of this reaction, the authors demonstrated the unique mechanism and broad substrate scope of the developed reaction, which will be of interest to the readership of this journal, especially researchers working in organic chemistry. I just have concerns about the frontier orbital analysis in Figure 4, where the percentage values of the LUMO coefficients are presented.

(1) I suggest the authors describe specifically how they obtained these values. If the coefficients on each carbon atom were simply squared and summed up, then the resultant values are correct

because the basis functions of 6-31G(d,p) have non-zero overlap.

(2) I also suggest plotting the LUMOs, so that the reader can get an idea of what they look like.

(3) According to the description in the SI, the alpha-LUMO was analyzed. Why did the authors use the alpha-LUMO instead of the beta-LUMO?

Our Responses to the Comments of the Reviewers

Reviewer 1

Comment 1: *This paper by Chao Feng and co-workers submitted to Nature Communications reports on the ring-opening of aryl cyclopropanes using a photocatalytic system. Numerous nucleophiles including alcohols, nitrogen-based nucleophiles, and (pseudo) halides are added in a 1,3-fashion. The proton is located next to the aryl residue. Moderate to good yields have been obtained. In contrast to common donor-acceptor cyclopropane chemistry the nucleophiles does not attack at the carbon next to the aryl, but at the other. A plausible mechanism based on the most reasonable experiments is provided. In general, this is a very nice report that heavily expands the ring-opening chemistry of cyclopropanes. I strongly support its publication in Nature Communications after some minor revisions noted below:*

Our response: We greatly thank the reviewer for his/her support of the publication of this work in *Nature Communications* after a minor revision. Accordingly, we have made every effort in revising the manuscript so that all the concerns of reviewers are addressed.

Comment 2: *The title is a bit misleading. I think the wording “sterically congested” is not necessary. However, I would definitely put “cyclopropane” in. What about “Photoredox-Catalyzed C-C Bond Cleavage of Cyclopropanes for the Formation of C(sp³)-Heteroatom Bonds”?*

Our response: We sincerely thank the reviewer for this valuable suggestion. As suggested by the reviewer, we have rephrased the title as “Photoredox-catalyzed C–C bond cleavage of cyclopropanes for the formation of C(sp³)–heteroatom bonds”.

Comment 3: *Although the introduction provides a broad overview of different methods, recent work on the electrochemically triggered ring-opening of cyclopropanes is missing. A sentence should be added, the respective references should be included: Angew. Chem. Int. Ed. 2021, 60, 15928; Org. Lett. 2021, 23, 5549.*

Our response: We sincerely thank the reviewer for this valuable suggestion. Accordingly, a sentence describing the mentioned work as “In addition, by the means of electrochemistry, Werz *et al.* have realized facile ring-opening functionalization of donor-acceptor cyclopropane/cyclobutane, which also traversed through radical cation intermediates.” has been added in the revised manuscript. Respective references have been added as ref. 52 and 53 in the revised manuscript.

Comment 4: *From my point of view the numbering/nomenclature of the compounds is very confusing. Why not using simple numbers and a,b, c etc.*

Our response: We sincerely thank the reviewer for this valuable suggestion. Accordingly, simple numbers have been used for numbering of compounds in the revised manuscript.

Comment 5: *Why do the authors state that their reactions schemes are figures? I think Scheme would be more appropriate.*

Our response: We thank the reviewer for this suggestion. However, according to the format requirement of Nature Communication, **schemes are not allowed and figures should be used instead**. Thus, our reactions are presented as figures.

Comment 6: *endo should be written in italics.*

Our response: We sincerely thank the reviewer for this valuable suggestion. The word “endo” has been written in italics in the revised manuscript.

Comment 7: *From my point of view it is not clear whether 3aS1 is a thiocyanate (RS-CN) or an isothiocyanate (R-N=C=S). Were the reported data compared with literature-known ones? It should be clearly elucidated which part of the amphiphilic nucleophile attacks.*

Our response: We sincerely thank the reviewer for this valuable suggestion. Recently, there are several reports in which trimethylsilyl isothiocyanate was used as nucleophile (*Org. Chem. Front.* **2021**, 8, 3076; *Org. Lett.* **2021**, 23, 4342; *Org. Chem. Front.* **2022**, 9, 2963; *Org. Lett.* **2022**, 24, 1742). In their work, the structures of products were all assigned as thiocyanates and were further unambiguously confirmed by single crystal X-ray diffraction analysis (*Org. Lett.* **2021**, 23, 4342, CCDC: **1817507**). We have also compared ¹³C NMR spectra of **3aS1** (compound **98** of the revised manuscript) and compounds of the aforementioned articles, a characteristic peak around 111 ppm was observed in all cases, which is indicative of a thiocyanate moiety. A similar structure with an isothiocyanate moiety has also been reported, with characteristic ¹³C NMR peak around 129 ppm (*Chem. Lett.* **2006**, 35, 1262). Taken together, we believe a thiocyanate is contained in **3aS1** (compound **98** of the revised manuscript).

Comment 8: *Mechanism: Are there any ideas of which is the rate-limiting step? An in-depth comparison on the kinetics with different cyclopropanes (as it was once made for donor-acceptor cyclopropanes: *Angew. Chem. Int. Ed.* 2019, 58, 1955) was not made and would exceed the scope of this study. Nevertheless, is there any information*

whether the kinetics is more determined by the type of cyclopropane or the type of nucleophile?

Our response: We sincerely thank the reviewer for this valuable suggestion. To gain better insight into the kinetic of the reaction, a set of competition reactions were executed. First, the influence of nucleophile was investigated by competition reaction of two arylcyclopropanes. The chloride ion exhibited much higher reactivity over trifluoromethanesulfonamide and acetic acid, two nucleophiles also employed in this work. This result indicated that the nucleophilicity of nucleophile had a significant impact on the reaction rate.

We have also compared reactivity of arylcyclopropanes with different alkyl substitution pattern by a similar competitive experiment. Substrate with two methyl substituents on the cyclopropane reacted faster than the counterpart with one methyl substituent upon reaction with ethanol, which is consistent with the trend we concluded in our manuscript, "...demanding substituents on cyclopropane are not detrimental but conducive to the ring-opening cleavage...".

Finally, the substitution effect on the aryl ring is surveyed by comparing relative reaction rate of structurally similar arylcyclopropanes with different aryl substituents. In these cases we choose chloride as the nucleophile because electron-poor arylcyclopropane reacted sluggishly with nucleophiles such as ethanol and TfNH₂. The reaction outcome showed that a strong electron-donating methoxy group could significantly accelerate the reaction rate to nearly one magnitude, whereas other weak electron-donating or electron-withdrawing groups reacted in similar rates, with no

obvious trend. Unfortunately, no Hammett plot could be fitted from the current data, which might be caused by the high nucleophilicity of chloride that minimize the effect of substituents with weak electron property. The extraordinary reaction rate of the methoxy substituted cyclopropane might be due to the relative lower oxidation potential of this particular substrate compared with the unsubstituted and *p*-bromide substituted ones (1.30 V vs 1.55 V and 1.60 V). From another point of view, the high reactivity of methoxy substituted cyclopropane could also be rationalized by more effective molecular interaction between SOMO of the radical cation intermediate and HOMO of the nucleophile (*Angew. Chem. Int. Ed.* **2012**, *51*, 7259; **2022**, *61*, e202206064).

In conclusion, three factors: nucleophilicity of nucleophile, alkyl substituent on the cyclopropane and substituent on the aryl ring have been demonstrated to significantly affect kinetic of the reaction. Unfortunately, we are unable to quantification the impact of each factor at current stage. As the three factors are separately involved in the initial single-electron-oxidation step and the ring-opening nucleophilic attack step, we are also unable to identify the rate-determining step for individual reactions. More research in depth about the kinetic of the reaction may help to gain insight into a deeper understanding of the mechanism. Results of the preliminary kinetic study has been added in the revised SI.

Comment 9: *Supporting Information: From my point of view, IR data are an important part of characterizing new compounds. These data should be added. I guess that also the question whether it is RSCN or RCNS might be easily solved with IR spectroscopy.*

Our response: We sincerely thank the reviewer for this valuable suggestion. Accordingly, IR data of new compounds were added in the revised Supporting Information.

Reviewer 2

Comment 1: *The manuscript by Feng and co-workers describes a photoredox/HAT catalysed hydrofunctionalisation of aryl cyclopropanes with a diverse range of heteroatom nucleophiles. Key to the success of the reaction is the single-electron oxidation of the aryl cyclopropanes into radical cations, which converts the cyclopropane into an electrophilic species that more closely matches the reactivity of donor–acceptor cyclopropanes. The authors have previously reported related nucleophilic ring-opening reactions of aryl cyclopropane radical cations (see references 47 and 53), however, in the current manuscript they demonstrate that the benzylic radical intermediates can be intercepted by a hydrogen atom transfer (HAT) with thiophenol, instead of trapping with oxygen or allyl sulfones. An impressive substrate scope is demonstrated for both the aryl cyclopropanes and nucleophiles, which include alcohols, triflamide, N-heterocycles, chloride and fluoride. The yields are generally high and the regioselectivities are excellent in nearly all cases. The authors also provide DFT calculations to explain the high regioselectivity observed for unsymmetrical 1,2-diarylcyclopropanes.*

*Although this is a nice piece of work, I do not think it represents a significant advance to the field. The methodology is a logical extension to the author’s (references 47 and 53), and Studer’s (references 56-57), previous work where they have replaced various radical traps with diphenyl disulfide to enable HAT. It appears that the authors have simply taken Nicewicz’s previously reported conditions and substrate scope for hydrofunctionalisations of aryl alkenes (reference 58; *J. Am. Chem. Soc.* 2012, 134, 18577; *J. Am. Chem. Soc.* 2013, 135, 9588; *J. Am. Chem. Soc.* 2013, 135, 10334; *Angew. Chem. Int. Ed.* 2014, 53, 6198; *Nat. Chem.* 2014, 6, 720) and applied them to aryl cyclopropanes. Indeed, the authors have already demonstrated that the transition from aryl alkenes (reference 49) to aryl cyclopropanes (reference 53) is readily achievable in fluoroallylation reactions. In addition, little further insight into the mechanism of the reaction has been provided (see reference 45-47 and 56-57).*

Based on this, I do not support publication in Nature Communications. The authors should consider the following points before resubmission to another journal.

Our response: We sincerely thank the reviewer for his/her effort and time to review this manuscript and for comments/suggestions his/her raised as listed below. However, we do not think the reviewer’s opinion about the novelty of our work is a fair comment. Our group is one of the leading groups in the area of photoredox ring-opening functionalization of arylcyclopropanes and this work is not a simple extension of our prior work (*Nat. Commun.* **2019**, 10, 4367; *Org. Lett.* **2020**, 22, 8681). The emphasis of our previous work is proof of concept that arylcyclopropane and gem-difluorocyclopropane could be activated under photoredox conditions to undergo ring-

opening functionalization upon nucleophilic attack, and the scope of nucleophiles are severely restricted to pyrazole and fluoride. In this work, we have proved that a much wider scope of nucleophile, from alcohol, carboxylic acid, sulfamide to chloride, trimethylsilyl azide and trimethylsilyl isothiocyanate, could be facily applied to this photoredox ring-opening reaction. The structural diversity and functional group compatibility have been demonstrated as showcased by late-stage functionalization of a set of natural product and pharmaceutical molecules. In addition, the current work has provided a rare solution to the stereospecific construction of quaternary carbon–heteroatom bond, which is difficult to accomplish by other methods. As for the second issue raised by the reviewer, though our current work shared somewhat similar conditions with a series of Nicewicz’s work, there are fundamental differences between these works. First, in Nicewicz’s work, the alkene is single-electron-oxidized to form an alkenyl radical cation intermediate. However, in our system, according to our experiment result and mechanistic study by others, the reaction proceeded via a ring-closed radical cation intermediate, which secured stereospecific transformation with enantioenriched substrates. **Second, by our method, various quaternary carbon–heteroatom bond could be readily forged, which is inaccessible, at least not demonstrated by Nicewicz’s methods.** We believe this difference is also caused by mechanism difference. **The synergistic effect of photocatalyzed SET-oxidation and strain-release, which is absent in Nicewicz’s examples, is rationalized to be the key for the fidelity of quaternary carbon–heteroatom formation.**

Taken together, this work is neither a simple extension of works of us and others nor a cyclopropane version of Nicewicz’s alkene hydrofunctionalization methodology. We hope that publication of this work in *Nature Communication* can stimulate more endeavor in this nascent yet promising area.

Comment 2: *The manuscript would benefit from careful proofreading, as there are numerous grammatical errors.*

Our response: We sincerely thank the reviewer for this valuable suggestion. We have checked the manuscript carefully before uploading the revised version.

Comment 3: *Error with H/R3 groups in Figure 1a.*

Our response: We sincerely thank the reviewer for pointing out this mistake. The error in figure 1a has been corrected in the revised manuscript.

Comment 4: *In Figure 1d, the reaction arrow showing the “intramolecular electron shift” structure to the “benzylic radical” is misleading because these are not separate species, they are resonance structures.*

Our response: We sincerely thank the reviewer for this valuable suggestion. Our initial intention was to illustrate the generation pathway of the benzylic radical intermediate. Indeed, the benzylic radical was produced directly upon ring-opening nucleophilic attack of the radical cation intermediate. Accordingly, the expression “intramolecular electron shift” has been removed from figure 1d and reaction arrow has been replaced by a double-headed arrow which is indicative of resonance structures.

Comment 5: $E_0^{1/2}$ should be defined or rewritten as $E_{1/2}$ or $E_{1/2}^{ox}$ to avoid confusion with standard reduction potential (E^0).

Our response: We sincerely thank the reviewer for pointing out this mistake. Accordingly, the expression of oxidation potential has been corrected in the revised manuscript.

Comment 6: “Diphenyl sulfide” should be “diphenyl disulfide”.

Our response: We sincerely thank the reviewer for pointing out this mistake. Accordingly, “diphenyl sulfide” has been corrected to “diphenyl disulfide” in the revised manuscript.

Comment 7: *The hydroaminations reported by Nicewicz (J. Am. Chem. Soc. 2013, 135, 9588; Angew. Chem. Int. Ed. 2014, 53, 6198) need to be cited because it appears that these are where the authors took their initial reaction conditions from.*

Our response: We sincerely thank the reviewer for this valuable suggestion. Accordingly, the mention work of Nicerwicz is added as reference 65 and 66 in the revised manuscript.

Comment 8: *Tertiary alcohols are a notable omission from the nucleophile scope. These should be investigated and the results included, even if the reactions fails.*

Our response: We sincerely thank the reviewer for this valuable suggestion. Tertiary alcohol is not a good nucleophile under our optimized conditions. Actually, tertiary alcohol could serve as a proton source when proton-free nucleophiles were employed (**3aN10**, **3aS1**, numbering **97** and **98** in the revised manuscript). A sentence “Unfortunately, tertiary alcohol is not amenable under current conditions, perhaps due to its low nucleophilicity and steric hindrance.” has been added in the revised manuscript.

Comment 9: *The failed reaction in the presence of TEMPO in Figure 2a is not necessarily indicative of a radical-engaged reaction. It likely that TEMPO simply reductively quenches the excited state photocatalyst, which prevents the formation of any alkyl radical intermediates, rather than functioning as a radical trap. Proof of a*

radical-engaged reaction would require the observation of a alkyl radical-TEMPO adduct.

Our response: We sincerely thank the reviewer for this valuable suggestion. Indeed, negative result of reaction with TEMPO as radical inhibitor is not a strong proof for a radical-involved mechanism. However, the result of the clock experiment strongly supported the generation of a benzylic radical (Figure 2b). Accordingly, the description of reaction as radical scavenger has been revised to “Secondly, addition of stoichiometric amount of TEMPO as radical inhibitor into the reaction of **1** and ethanol led to complete inhibition of product formation (Fig. 2a)”.

Comment 10: *Line 264: “During the investigation, the unusual regioselectivities in some cases have attracted our attention (3tO1, 3uO1, 3tN2, 3uN1, 3uN2, 3adO1) – should it be 3ahO1?”*

Our response: We sincerely thank the reviewer for pointing out this mistake. 3adO1 is a typo of 3ahO1 and the numbering “3tO1, 3uO1, 3tN2, 3uN1, 3uN2, 3ahO1” has been changed to “**19, 20, 43, 44, 45, 115**” in the revised manuscript.

Comment 11: *Line 265: “Results of previous mechanistic studies that the nucleophilic substitution tends to produce the more stable radical intermediate, i.e. substitution occur at the β position of the electron-rich aryl ring, was in contradiction with the reaction outcome.” – the reaction outcomes do not contradict this because benzylic radicals are always generated. Also, this statement is not relevant to the calculations in figure 4, which only include the LUMO coefficients of the radical cation, not the stability of the radical formed after nucleophilic attack.*

Our response: We sincerely thank the reviewer for pointing out this issue and feel sorry that we haven’t described it clearer. In order to avoid misleading, the sentence “Results of previous mechanistic studies ... in contradiction with the reaction outcome.” was removed in the revised manuscript. While producing radical intermediate with more stability from the thermodynamic consideration would influence the regioselectivity of nucleophilic attack, the regioselectivity of reactions, especially for diaryl cyclopropane derivatives that produce radical intermediate with similar stabilities, is mainly subjected to orbital control manner which entails more effective orbital overlap during HOMO/LUMO interaction in the transition state. The outcomes of reactions are all in accordance with our calculations indicating that the carbon atom with largest LUMO coefficient is attacked by external nucleophiles.

Comment 12: *Line 282: “acridine photocatalyst” should be changes to “acridinium photocatalyst”.*

Our response: We sincerely thank the reviewer for pointing out this mistake. “acridine” has been corrected to “acridinium” in the revised manuscript.

Comment 13: *The title of reference 15 is wrong.*

Our response: We sincerely thank the reviewer for pointing out this mistake. The title of reference 15 (ref. 12 of the revised manuscript) has been corrected in the revised manuscript.

Reviewer 3

Comment 1: *The authors present a novel type of photoredox-coupled C(sp³)-heteroatom bond formation reaction, which involves a single electron transfer oxidation of the aryl group and C-C bond cleavage. Although the involvement of iridium may not be a desirable feature of this reaction, the authors demonstrated the unique mechanism and broad substrate scope of the developed reaction, which will be of interest to the readership of this journal, especially researchers working in organic chemistry.*

Our response: We greatly thank the reviewer for his/her support of the publication of this work in *Nature Communications* after a minor revision. Accordingly, we have made every effort in revising the manuscript so that all the concerns of reviewers are addressed.

Comment 2: *I suggest the authors describe specifically how they obtained these values. If the coefficients on each carbon atom were simply squared and summed up, then the resultant values are correct because the basis functions of 6-31G(d,p) have non-zero overlap.*

Our response: We sincerely thank the reviewer for this valuable suggestion. After optimizing the structure, we obtained its wavefunction and the electronic structure analysis was performed with the Multiwfn (Multifunctional Wavefunction Analyzer) including orbital composition analysis with Mulliken partition. VMD (Visual Molecular Dynamics) software was used to render the isosurface maps of various orbitals based on the files exported by Multiwfn.

Comment 3: *I also suggest plotting the LUMOs, so that the reader can get an idea of what they look like.*

Our response: We sincerely thank the reviewer for this valuable suggestion. Accordingly, we have plotted the LUMOs of the calculated radical cation intermediates and these plots have been added in the revised manuscript and SI.

Comment 4: According to the description in the SI, the alpha-LUMO was analyzed. Why did the authors use the alpha-LUMO instead of the beta-LUMO?

Our response: We sincerely thank the reviewer for this question. According to the calculation, the energy of alpha-LUMO is higher compared with that of beta-LUMO. In addition, the alpha-LUMO is of antibonding character with respect to the C–C bond being cleaved, which engenders more effective orbital interaction with the HOMO of nucleophile. Therefore, the alpha-LUMO was analyzed for nucleophilic C–C bond cleavage.

REVIEWERS' COMMENTS

Reviewer #1 (Remarks to the Author):

The authors have done a very careful job in improving the quality of the paper. Of course, I had a major look on the criticism I gave in my previous report. From my point of view every point was clarified. The title is much better now. Missing references were included. Congratulation to this nice piece of work!

Reviewer #3 (Remarks to the Author):

The authors have addressed the raised issues appropriately.

REVIEWERS' COMMENTS

Reviewer #1:

Comment 1: *The authors have done a very careful job in improving the quality of the paper. Of course, I had a major look on the criticism I gave in my previous report. From my point of view every point was clarified. The title is much better now. Missing references were included. Congratulation to this nice piece of work!*

Our response: We greatly thank the reviewer for his/her support of the publication of this work in *Nature Communications*.

Reviewer #3:

Comment 1: *The authors have addressed the raised issues appropriately.*

Our response: We greatly thank the reviewer for his/her support of the publication of this work in *Nature Communications*.